# Federated Spectral Graph Transformers Meet Neural Ordinary Differential Equations for Non-IID Graphs

**Kishan Gurumurthy**[*]                                              *kishangurumurthy@gmail.com*
*Department of Computer Science*
*Indian Institute of Information Technology Kottayam (IIITK)*

**Himanshu Pal**[*]                                              *himanshu.pal@research.iiit.ac.in*
*Machine Learning Lab*
*International Institute of Information Technology Hyderabad((IIITH)*

**Charu Sharma**                                              *charu.sharma@iiit.ac.in*
*Machine Learning Lab*
*International Institute of Information Technology Hyderabad (IIITH)*

**Reviewed on OpenReview:** *https: // openreview. net/ forum? id= TR6iUG8i6Z*

## Abstract

Graph Neural Network (GNN) research is rapidly advancing due to GNNs' capacity to learn distributed representations from graph-structured data. However, centralizing large volumes of real-world graph data for GNN training is often impractical due to privacy concerns, regulatory restrictions, and commercial competition. Federated learning (FL), a distributed learning paradigm, offers a solution by preserving data privacy with collaborative model training. Despite progress in training huge vision and language models, federated learning for GNNs remains underexplored. To address this challenge, we present a novel method for federated learning on GNNs based on spectral GNNs equipped with neural ordinary differential equations (ODE) for better information capture, showing promising results across both homophilic and heterophilic graphs. Our approach effectively handles non-Independent and Identically Distributed (non-IID) data, while also achieving performance comparable to existing methods that only operate on IID data. It is designed to be privacy-preserving and bandwidth-optimized, making it suitable for real-world applications such as social network analysis, recommendation systems, and fraud detection, which often involve complex, non-IID, and heterophilic graph structures. Our results in the area of federated learning on non-IID heterophilic graphs demonstrate significant improvements, while also achieving better performance on homophilic graphs. This work highlights the potential of federated learning in diverse and challenging graph settings. Open-source code available on GitHub.[1]

## 1 Introduction

Graph Neural Networks (GNNs) (Scarselli et al., 2008) have emerged as powerful tools for learning representations from graph-structured data, which is prevalent in various domains such as social networks, molecular biology, and traffic systems (He et al., 2021; Liu et al., 2024a). In these applications, data is often decentralized and subject to privacy concerns, making traditional centralized approaches to training GNNs less viable due to data privacy and regulatory restrictions (He et al., 2021; Ju et al., 2024). Federated Learning (FL) offers a promising solution by enabling decentralized learning across multiple clients without sharing raw data, thereby preserving privacy (Zhang et al., 2023; Wu et al., 2022). However, applying FL to graph data introduces unique challenges due to the non-Independent and Identically Distributed (non-IID)

---

[*]These authors contributed equally to this work.
[1]https://github.com/SpringWiz11/Fed-GNODEFormer

nature of graph data and the heterophilic relationships often present in real-world graphs. This has led to the development of Federated Graph Neural Networks (FedGNNs), a novel area of research that seeks to leverage the strengths of FL to address these challenges (Liu et al., 2024a) in GNNs.

Federated Learning (FL) was introduced by McMahan et al. (2017) as a method for training machine learning models across decentralized data sources while maintaining data privacy (Zhang et al., 2023). FL has been successfully applied in domains such as image and language processing, where data is typically IID. However, its application to graph-structured data presents unique challenges because graph data is inherently non-IID and can involve complex dependencies and relationships (Zhang et al., 2023). GNNs are well-suited for modeling such relationships in graph data and have been extensively used for tasks like node classification, link prediction, graph clustering and classifications (Fu et al., 2022). The integration of FL with GNNs, therefore, aims to leverage the advantages of both, enabling privacy-preserving, decentralized learning on graph data while addressing the complexities associated with non-IID and heterophilic graphs (Liu et al., 2024a). Despite its potential, the development of FedGNNs that effectively handle these challenges remains relatively under-explored.

The non-IID nature of graph data complicates the training process in a federated setting, as traditional FL algorithms are designed for IID data and may not perform optimally with graph data (He et al., 2021; Zhang et al., 2023). Real-world applications often involve complex graph structures where data points are non-IID, and relationships are heterophilic. Traditional FL methods may struggle with convergence, communication overhead, and privacy risks in these settings. Therefore, there is a need for efficient solutions that can handle such complexities effectively while preserving privacy and optimizing bandwidth usage.

This work addresses the challenges of federated learning on graph-structured data, particularly focusing on non-IID scenarios involving both homophilic and heterophilic graphs. We propose a method that leverages spectral GNNs and neural ODE (Xhonneux et al., 2020; Yin et al., 2024; Rusch et al., 2022) to enhance the capture of complex graph relationships and dynamics. Our approach demonstrates superior performance in non-IID settings for both graph types, while also maintaining competitive results on IID data compared to state-of-the-art methods. It focuses on optimizing communication costs and preserving data privacy, making it well-suited for real-world federated learning scenarios. The design ensures privacy preservation and bandwidth efficiency, which are important for practical applications where complex, non-IID graph structures are common. Our initial motivation stemmed from these challenges, which led us to explore how GNODEFormer could be adapted to work effectively in federated settings. Notably, while our approach is tailored for federated learning, its design is versatile enough to perform well even in centralized settings, showcasing its broad applicability to various graph learning scenarios. Key contributions of our work include:

- We propose a novel GNN architecture for centralized settings, called GNODEFormer.

- We extend this architecture to a FL setting, introducing Fed-GNODEFormer, designed to handle both homophilic and heterophilic graphs under non-IID data conditions.

- Our empirical results demonstrate enhanced performance on non-IID data and competitive results on IID data when compared to state-of-the-art methods.

- We develop privacy-preserving and bandwidth-efficient strategies to enable federated learning for graph-structured data across various types and data distributions (see A).

The remainder of this work is structured as follows: Section 2 reviews related work on federated learning and GNNs, identifying gaps that our approach addresses. Section 3 details our proposed method, explaining the use of spectral GNNs and neural ordinary differential equations. Section 4 outlines the experimental setup, detailing the datasets and evaluation metrics, presents the results by comparing our method's performance with existing approaches, and discusses the implications of our findings along with the limitations of the study. Finally, Section 5 concludes the paper and suggests potential directions for future research.

## 2 Related Work

Data privacy and control have become significant concerns in modern machine learning, particularly due to the vast amounts of data required for deep learning (Shokri & Shmatikov, 2015; Albrecht, 2016). Recent works (Ju et al., 2024) highlights key limitations in deploying GNNs under practical conditions, including class imbalance, noisy labels, privacy constraints, and out-of-distribution (OOD) generalization. Federated learning (FL) has emerged as an innovative solution, allowing decentralized model training across multiple devices while preserving data privacy. Simultaneously, Graph Neural Networks (GNNs) have gained prominence for their ability to model complex relational data. The combination of these fields, Federated Learning in Graph Neural Networks (FL-GNN), offers both new opportunities and challenges in addressing data privacy concerns while effectively modeling heterogeneous graph data.

GNNs are categorized into message-passing neural networks (MPNNs) and spectral-based GNNs (Ortega et al., 2018; Dong et al., 2020; Wu et al., 2019; Zhu et al., 2021). MPNNs, like Graph Convolutional Networks (GCNs) (Kipf & Welling, 2017) and GraphSAGE (Hamilton et al., 2017a), propagate information between neighboring nodes. Graph Attention Networks (GAT) (Veličković et al., 2018) extended this by using attention mechanisms to dynamically weight neighboring nodes, further refining the learning of node representations. In contrast, Spectral GNNs use the graph Laplacian's spectral properties in the frequency domain (Singh & Chen, 2023). GCNs are also classified as spectral methods, employing a simplified spectral convolution. ChebNet (Defferrard et al., 2016) uses Chebyshev polynomials to approximate the graph Fourier transform, which reduces computational demands.

Recent advancements like Specformer (Bo et al., 2023) have integrated spectral GNNs with transformer architectures to effectively capture both local and global graph characteristics. Specformer leverages the standard transformer encoder design, which, while effective, may fall short in modelling the continuous feature transformation (information propagation) across graph structures. Our approach addresses this through higher-order Ordinary Differential Equations (ODEs). In contrast to conventional transformers that model information flow using distinct layers, Neural ODEs offer a more fluid and mathematically solid depiction of continuous processes, as recent interpretations suggest that transformers can be viewed as discrete Neural ODEs (Hashimoto et al., 2024). Building on this insight, our approach incorporates higher-order Ordinary Differential Equations (ODEs), such as second- and fourth-order Runge-Kutta methods, to capture richer dynamics in feature propagation. By utilizing Neural ODEs, our model demonstrates improved adaptability to heterophilic and extreme non-IID settings, where traditional transformers often struggle.

Recent advancements have explored the integration of continuous-time dynamics into graph learning. Zhuang et al. (2020) models node evolution using neural ordinary differential equations, while PDE-GCN (Eliasof et al., 2021) draws inspiration from partial differential equations to improve feature propagation across graphs. These models demonstrate the power of continuous formulations in enhancing representation learning but are developed for centralized settings and do not consider distributional heterogeneity. Complementing this, Ju et al. (2024) provide a comprehensive survey of practical challenges in GNN deployment, such as class imbalance, noise, privacy, and out-of-distribution generalization. Our work builds on these directions by integrating a neural ODE mechanism within a federated GNN architecture—explicitly addressing the challenges of non-IID data and privacy in real-world settings, which remain underexplored in prior differential-equation-based GNNs.

A first-order ordinary differential equation (ODE) problem is defined by an equation involving a first-order derivative and an initial condition. These equations often lack closed-form solutions and are typically solved using numerical methods. Euler's method, the simplest numerical ODE solver, discretizes the time derivative via a first-order approximation (Ascher & Petzold, 1998). This temporal discretization forms the basis of residual connections/networks, as first proposed by (Weinan, 2017). Subsequent works, such as (Lu et al., 2018; Chen et al., 2018; Zhu et al., 2022; Eliasof et al., 2021), demonstrated that any parametric ODE solver can be interpreted as a deep residual network with infinite layers, optimized through backpropagation (Rumelhart et al., 1986).

Our work aligns closely with (Li et al., 2021), which explores connections between transformers and ODEs. They propose that a Transformer's residual block can be viewed as a higher-order ODE solver, leading

to the ODE Transformer architecture inspired by the Runge-Kutta method. Similarly, (Lu et al., 2019) also interprets transformers as numerical ODE solvers for convection-diffusion equations in multi-particle systems, leading to architectural modifications such as relocating self-attention layers. However, (Li et al., 2021) highlights performance degradation due to error accumulation in stacked first-order ODE blocks and addresses this by introducing high-order blocks, resulting in notable BLEU score improvements.

FL has progressed significantly in recent times. McMahan et al.'s work on Federated Averaging (FedAvg) (McMahan et al., 2017) established the field's foundation, addressing privacy issues and reducing centralized data storage needs. Recent research has tackled distributed data privacy through methods like DARLS for learning causal graphs from distributed data (Ye et al., 2024). FL has emerged as a privacy-preserving solution for on-device learning, balancing communication costs and model accuracy (Pfeiffer et al., 2023). Researchers have also developed FL algorithms for data streams, improving performance on various machine learning tasks (Marfoq et al., 2023). For heterogeneous edge computing, efficient mechanisms like FedCH (Wang et al., 2023) have been introduced, reducing completion time and network traffic. These studies (Liu et al., 2024a) collectively advance FL's applicability in privacy-preserving collaborative learning and distributed model training.

Combining FL with GNNs (FL-GNN) creates a novel paradigm that uses the strengths of both fields. GNNs are good at handling graph data, common in areas like social networks and biology (Hamilton et al., 2017b). However, centralized training of GNNs raises concerns about data privacy (Albrecht, 2016; Cui et al., 2024), regulatory compliance, and logistical issues, particularly when handling sensitive information (Liu et al., 2024c). Federated learning helps by allowing decentralized training while keeping data local. FedGraphNN (He et al., 2021) is an FL-GNN method that helps train GNNs in a federated way and provides a way to test different GNN models and datasets. It shows the challenges of federated training with non-IID data, where data differs across clients. Standard FL methods like FedAvg may not work well for GNNs, suggesting a need for better techniques. Federated Graph Convolutional Networks (FedGCNs) (Yao et al., 2023) apply Graph Convolutional Networks to federated learning. They handle graph data challenges through local training followed by global updates (Liu et al., 2024b). This improves privacy and works well with large datasets. However, FedGCNs often perform poorly with extreme non-IID data and struggle to combine updates from different graph structures. Thus, our aim is to build a method that addresses issues of both heterophilic graphs and non-IID data. Our work aligns with this direction by explicitly addressing two of these challenges: non-IID data distributions and privacy preservation in federated settings. Through the design of GNODEFormer and its federated extension, we contribute towards building GNN architectures that are both robust to distributional heterogeneity and mindful of privacy constraints in real-world deployments.

## 3 Our Method

In this section, we introduce our proposed model, a novel neural network architecture that integrates a transformer-based framework (Vaswani, 2017) with Ordinary Differential Equations (ODEs) (Runge, 1895; Kutta, 1901; Butcher, 1996; Ascher & Petzold, 1998) for node classification on both homophilic and heterophilic graphs. We begin by presenting preliminary information on semi-supervised node classification in centralized and federated settings, emphasizing the importance of capturing dependencies to generate meaningful representations.

Let $\mathcal{G} = (\mathcal{V}, \mathcal{E})$ represent a global graph dataset, where $\mathcal{V}$ is the set of nodes with $|\mathcal{V}| = n$, and $\mathcal{E}$ is the set of edges. For this work, we assume that $\mathcal{G}$ is undirected. The graph is associated with an adjacency matrix $\mathcal{A} \in {0, 1}^{n \times n}$, where $\mathcal{A}ij = 1$ if an edge exists between node $i$ and node $j$, and $\mathcal{A}ij = 0$ otherwise. In semi-supervised node classification, a subset of nodes, $k \subset \mathcal{V}$, is labeled, while the remaining nodes lack labels. The objective is to predict the labels of these unlabeled nodes using a trained model. In our case, this model is GNODEFormer (Graph-Neural-Ordinary-Differential-Equation-Former), as described in Section 3.1.

To effectively model and analyze graph-structured data, the graph Laplacian plays a crucial role. In Graph Signal Processing (GSP), the graph Laplacian is defined as $\mathcal{L} = \mathcal{D} - \mathcal{A}$, where $\mathcal{D}$ is the diagonal degree matrix with entries $\mathcal{D}ii = \sum j \mathcal{A}ij$ for all $i \in \mathcal{V}$, and $\mathcal{D}ij = 0$ for $i \neq j$. Similarly, the normalized graph Laplacian is expressed as $\mathcal{L} = \mathcal{I}_n - \mathcal{D}^{-\frac{1}{2}} \mathcal{A} \mathcal{D}^{-\frac{1}{2}}$, where $\mathcal{I}_n$ denotes the $n \times n$ identity matrix.

In Section 3.2, we extend GNODEFormer to the Federated Learning scenario. Specifically, we begin by partitioning the global graph dataset, $\mathcal{G}$, into **m** disjoint subgraphs, $\mathcal{G}i = (\mathcal{V}i, \mathcal{E}_i)$. To achieve a diverse and representative distribution of nodes and edges across the **m** subgraphs, we employ a Dirichlet distribution during the partitioning process. Each subgraph is then assigned to a unique client, ensuring that the partitions remain disjoint.

## 3.1 GNODEFormer

In this section, we introduce our proposed model, Graph-Neural-Ordinary-Differential-Equation-transFormer (GNODEFormer), and discuss its components and functioning. GNODEFormer is a centralized architecture designed to capture continuous data dynamics and retain complex dependencies in graph-based learning tasks. The model leverages ODE-inspired mechanisms, incorporating second-order and fourth-order Runge-Kutta methods (Li et al., 2022) to refine the outputs of transformer layers. These mechanisms enable GNODEFormer to model the temporal and structural evolution of graph data effectively.

To further enhance learning, we utilize the Residual Layer History mechanism, which accumulates and normalizes intermediate layer outputs. This mechanism allows the model to retain crucial information across layers, improving its ability to capture intricate dependencies within the graph structure.

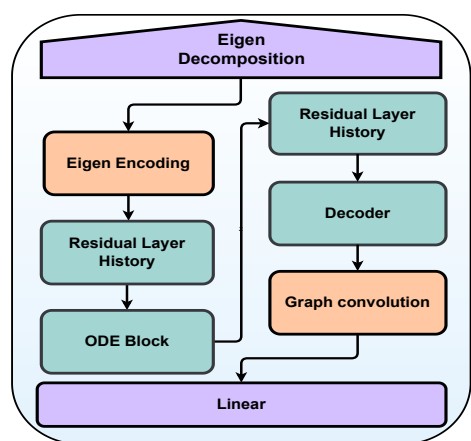

Finally, the decoder reconstructs the graph Laplacian matrix from the refined representations. This enables the model to perform spectral graph convolution, ensuring that the learned embeddings align with the graph's spectral properties and effectively encode its structure.

By keeping its essential elements—Eigen Decomposition and Eigenencoding for spectral analysis—GNODEFormer expands upon Specformer. In order to better handle complex graph relationships and heterophilic/non-IID data, it introduces Neural ODE layers for continuous transformations, a Residual Layer History mechanism for normalizing outputs, and enhanced convolution that combines Neural ODEs with spectral methods.

Figure 1: GNODEFormer architecture: Integrating ODE-inspired Runge-Kutta methods, Residual Layer History, and spectral graph convolution for enhanced graph representation learning.

### 3.1.1 Eigen Decomposition

Eigen decomposition plays a crucial role in analyzing the graph Laplacian spectra, which provides valuable frequency-related insights into the structure of the graph. Specifically, the *normalized graph Laplacian* $\mathcal{L}$ is a real symmetric matrix, and its eigen decomposition can be expressed as:

$$\mathcal{L} = \mathcal{U}\Psi\mathcal{U}^T$$

Here, $\mathcal{U}$ contains the eigenvectors of $\mathcal{L}$, and $\Psi = \text{diag}([\gamma_1, \gamma_2, \gamma_3, ..., \gamma_n])$ is the diagonal matrix of corresponding eigenvalues, which lie in the range $[0, 2]$. The eigenvectors in $\mathcal{U} = [u_1, u_2, u_3, \ldots, u_n]$ are orthonormal, meaning that each eigenvector $u_j$ has a unit norm ($||u_j|| = 1$), and the dot product $u_i^T u_j = \delta_{ij}$, where $\delta_{ij}$ is the Kronecker delta function, equal to 1 if $i = j$ and 0 otherwise. This orthonormality is essential for preserving the graph's structural information during spectral transformations.

### 3.1.2 Eigenvalue Encoding

We leverage the transformer mechanism to design a powerful set-to-set spectral filter that utilizes both the magnitudes and relative differences of the eigenvalues. To overcome the limitations of using scalar eigenvalues directly, which would constrain the expressiveness of self-attention, we use the eigenvalue encoding functions 1 and 2 to transform each scalar eigenvalue into a rich vector representation, $E(\gamma) : \mathbb{R}^1 \rightarrow \mathbb{R}^d$.

The eigenvalue encoding functions are defined as: -

$$E(\gamma, 2i) = \sin\left(\frac{\epsilon \times \gamma}{10000^{2i/d}}\right) \tag{1}$$

$$E(\gamma, 2i+1) = \cos\left(\frac{\epsilon \times \gamma}{10000^{2i/d}}\right) \tag{2}$$

In the above equations, $i$ is the dimension of the vector representation and $\epsilon$ is a hyperparameter. Although the eigenvalue encoding (EE) is similar to the positional encoding (PE) of Transformer, they act quite differently. PE describes the information of discrete positions in the spatial domain. This motivates us to use the Ordinary Differential Equation to construct the further experiments. Eigenvalue encoding identifies the changes in frequency between eigenvalues and generates vector and multi-scale representations. These representations provide important insights into the eigenvalues within the spectral domain. Then, the eigenvalues are concatenated with their eigen encoding, $\mathcal{Z} = [\gamma_1||\rho(\gamma_1), \gamma_2||\rho(\gamma_2), ..., \gamma_n||\rho(\gamma_n)]^T \in \mathbb{R}^{n \times (d+1)}$. The result is then passed onto the ODE Transformer block which learns the dependency between the eigenvalues.

### 3.1.3 Ordinary Differential Equations and Layer History Mechanism

Our method incorporates ODE-inspired layers within its architecture, specifically utilizing Runge-Kutta integration methods such as RK2 and RK4 (Li et al., 2022). These methods iteratively refine the outputs of the network's layers, enabling continuous refinement of the model. This process enhances the model's ability to capture the dynamics of data flow, leading to more accurate and stable representations of complex data patterns.

The ODE integration process within each transformer layer can be described by

$$z^{l+1} = z^l + \sum_{i=1}^{p} w_i f_\theta\left(x_n + \sum_{j=1}^{i-1} a_{ij} k_j\right) \tag{3}$$

where $z^{(l)}$ is the input to the layer, $z^{(l+1)}$ is the output of the layer, $p$ is the order of the Runge-Kutta method (2, 3, or 4), $f_\theta$ represents the neural network layer function with parameters $\theta$, $w_i$ are the weights for combining each $k_i$ term, $a_{ij}$ are the coefficients for the intermediate steps, $k_i$ are intermediate values representing estimates of the change in $x$ over a step, and $k_j$ are the previously calculated $k$ values used in computing the current $k_i$.

This process refines the layer outputs to capture more subtle patterns. Following this, the model introduces the Residual Layer History mechanism, which deviates from conventional transformer models. Instead of passing the outputs directly between layers, the model accumulates and normalizes the outputs across layers, preserving a history of previous states. This accumulated history is then used to iteratively compute the model's next state, allowing for more nuanced feature representations. The residual update is given by:-

$$x_n = x_{n-1} + y_{n-1} \tag{4}$$

where $x_n$ represents the output at the current layer $n$, $x_{n-1}$ is the output from the previous layer $n-1$, and $y_{n-1}$ is the residual term added at layer $n-1$.

The use of Neural ODEs in Fed-GNODEFormer is motivated by their ability to model continuous feature transformation(information propagation) across graph structures. Unlike transformers, which approximate this process in discrete layers, Neural ODEs offer a principled framework for continuous modelling, as supported by recent interpretations of transformers as discrete Neural ODEs (Hashimoto et al., 2024). This continuous framework enables enhanced representation learning, particularly for non-IID and heterophilic graphs, by seamlessly integrating with spectral GNNs. Additionally, Neural ODEs' adaptive solver strategies improve computational efficiency, addressing the memory and processing challenges typical of federated graph learning frameworks.

### 3.1.4 Decoding and Graph Convolution

The representations and the continuous dynamics of the spectra, learned in the ODE stage, are passed to the decoder to predict new eigenvalues, denoted as $\gamma_n = \text{Decoder}(Z)$. These eigenvalues, generated by the ODE-Transformer block, are refined through a spectral filtering mechanism. Using these refined eigenvalues, the individual bases of the graph Laplacian are reconstructed as $\mathcal{L}_{\text{new}} = \mathcal{U}\text{diag}(\gamma_n)\mathcal{U}^T$.

The reconstructed $\mathcal{L}_{\text{new}}$ is then processed through fully connected layers equipped with residual connections. These layers, along with self-attention and feed-forward sub-layers, produce a new learnable base, $\hat{\mathcal{L}}$. This base is used to assign a separate graph Laplacian matrix to each feature dimension, enabling the model to perform spectral graph convolution effectively.

### 3.2 Fed-GNODEFormer

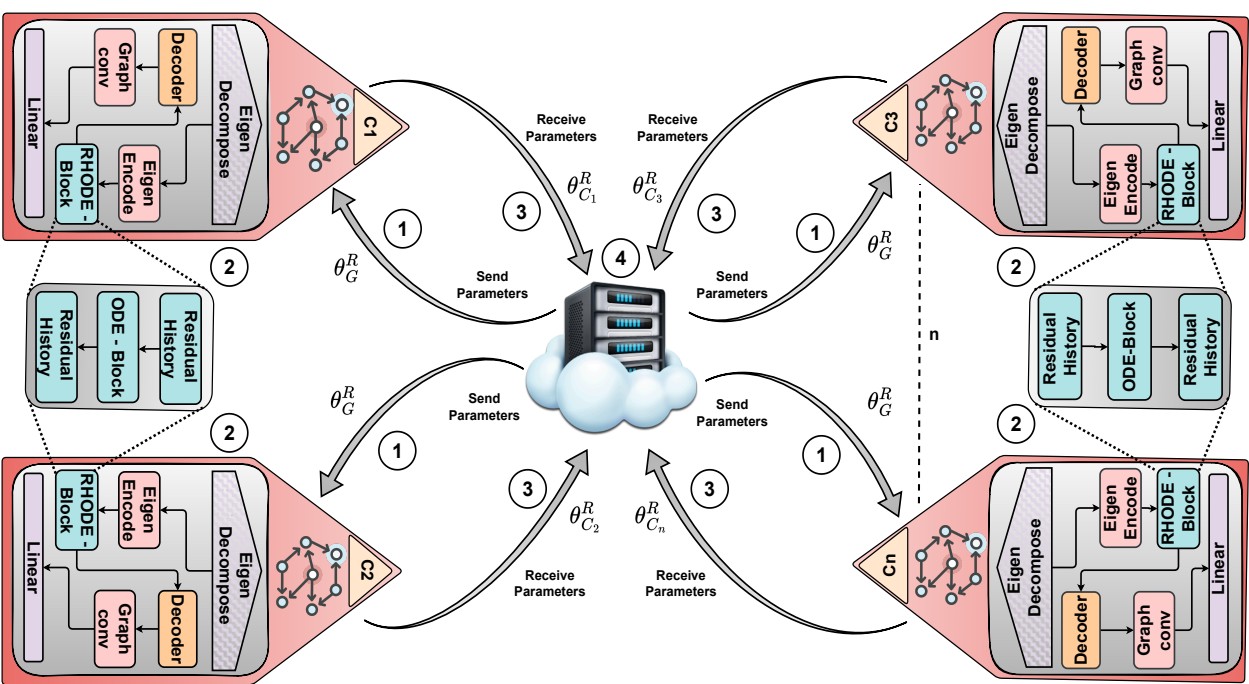

Figure 2: Fed-GNODEFormer: GNODEFormer in a Federated Learning setup. The diagram illustrates multiple clients, each with a local model, interacting with a central server. The numbered arrows represent the standard FL workflow: (1) the server sends the global parameters, (2) clients perform local updates, (3) clients send the updated parameters back, and (4) the server aggregates them. **Arrows indicate the flow of parameters and updates for explanation purposes and do not imply a directed graph structure.**

In this section, we introduce our Federated Graph Neural Ordinary Differential Equation transFormer, abbreviated as **Fed-GNODEFormer**, and discuss the performance of GNODEFormer in a decentralized machine learning scenario.

The global graph $\mathcal{G}$ is partitioned into **m** disjoint sub-graphs, $\mathcal{G}_i = (\mathcal{V}_i, \mathcal{E}_i)$, which are then allocated to a set of clients, $\mathcal{C} = \{c_1, c_2, c_3, \ldots, c_n\}$. The goal of Federated Learning (FL) is to minimize the global loss function across all clients while keeping the graph data decentralized. Each client $c_i \in \mathcal{C}$ trains GNODEFormer locally on its assigned sub-graph, $\mathcal{G}_i$.

A significant challenge in FL arises from the non-iid nature of the data, as the graph partitions across clients are often imbalanced or skewed. This heterogeneity increases the complexity of optimizing the global loss function compared to centralized learning. Due to this imbalance, clients may converge to local minima, where the model performs well on the client's local data but poorly on the global task. These local minima

can vary significantly between clients, particularly when the data distribution exhibits a high degree of non-iid-ness.

The global optimization task in FL is to aggregate these locally trained models effectively to minimize the global loss function. GNODEFormer demonstrates robust performance in such heterogeneous scenarios, excelling even when the data concentration parameter ($\alpha$) of the Dirichlet Distribution is as low as 0.01, indicating a high degree of non-iid-ness.

### 3.2.1 Federated Training

Subgraph sampling for federated training involves partitioning the global graph $G$ into $m$ subgraphs, $G_i = (V_i, E_i)$, which are then distributed among a set of clients $\mathcal{C}$. This partitioning uses a Dirichlet distribution to ensure a diverse and representative distribution of nodes and edges, effectively simulating *non-IID data conditions*. After partitioning of the graph into various subgraphs $\mathbf{m}$ and assigning them to the client set $\mathcal{C}$. For each client $c_i \in \mathcal{C}$, we now proceed to perform the eigen decomposition on the adjacency $\mathcal{A}_{c_i}$ of the local subgraph. This step plays a crucial role of capturing information in the high non-iid local graph $\mathbf{m}_i$ of the client $c_i$. Post the eigen decomposition step, each client will have its own eigenvalue set, $\gamma_{m_i}$ and eigenvectors $\mathcal{U}_{m_i}$. This can be analyzed as:

$$\mathcal{L}_{c_i} = \mathcal{U}_{m_i} \psi \mathcal{U}_{m_i}^T, \quad \text{where } \psi = \text{diag}([\gamma_{m_{i_1}}, \gamma_{m_{i_2}}, \gamma_{m_{i_3}}, \dots, \gamma_{m_{i_n}}]). \tag{5}$$

Next, we perform the eigen encoding using (1) and (2) to capture the relative frequency shifts of eigenvalues and provide high-dimensional vector representations.

$$E(\gamma_m, 2i) = \sin\left(\frac{\epsilon \gamma_m}{10000^{2i/d}}\right),$$
$$E(\gamma_m, 2i+1) = \cos\left(\frac{\epsilon \gamma_m}{10000^{2i/d}}\right). \tag{6}$$

After this step, the eigen values are concatenated with their eigen encoding $\mathcal{Z}_m = [\gamma_{m_1}||\rho(\gamma_{m_1}), \gamma_{m_2}||\rho(\gamma_{m_2}), \dots, \gamma_{m_n}||\rho(\gamma_{m_n})]^T$, where $\gamma_{m_i}$ represents the $i^{th}$ clients ($c_i$) local eigenvalue belonging to the local eigen set $\gamma_m$. The result is then passed on to the ODE inspired Transformer model to learn the dependencies between the eigen values. Here, we use the High-Order ODE solvers like Runge-Kutta methods to capture the continuous dynamics in the spectral domain. We taken second-order Runge-Kutta and fourth-order Runge-Kutta methods into consideration. In addition to this, we are also maintaining the residual layer history, as in, we preserve the history of the previous states. This result is then used to iteratively compute the model's next state. Then the result is passed onto the decoder block to filter the new eigenvalues, $\gamma_{nm_i}$ and learn them. This new eigen are then used to further enhance the model's ability to capture the spectral characteristic of the local graph. Then we go about reconstructing the individual bases $\mathcal{L}_{nc_i} = \mathcal{U}_{m_i} diag(\gamma_{nm_i})\mathcal{U}_{m_i}^T$. Finally, we perform spectral graph convolution. The spectral graph convolution is written as $f *_g x = \mathcal{U}_{m_i} g_\theta(diag(\gamma_{m_i}))\mathcal{U}_{m_i}^T x$. This result is then passed onto the fully connected layers equipped with local residual connections to construct a new learnable base $\hat{\mathcal{L}_m}$. This enables the local model to effectively learn the node representations.

The server is first assigned with $w_{global}$ and transmitted to all the participating clients. This feature of controlling the participating clients is controlled by a parameter called fraction fit. This can be viewed as $\mathcal{S} = \mathcal{C}.\mathcal{K}$, where $\mathcal{C}$ is the client set and $\mathcal{K}$ is the fraction fit. The participating client divides the sub-graph $m_i$ into the respective training set and testing set. It can be analysed as $\mathbf{m}_i^{train} = (\mathbf{v}_i^{train}, \mathbf{e}_i^{train})$ and $\mathbf{m}_i^{test} = (\mathbf{v}_i^{test}, \mathbf{e}_i^{test})$. The objective is to generate the labels to the unlabeled vertices. The clients undergo their own local model training process using the GNODEFormer on the local sub-graph $\mathbf{m}_i$ to minimize the local loss. Once the local model training process is completed, the weights from the participating clients $\mathcal{S}$ is sent back to the model to perform the server aggregation process. For server aggregation we choose to go with the standard FederatedAveraging algorithm, where all the data from the participating client set $\mathcal{S}$ undergoes weighted averaging. Here, the weights are proportional to the number of data points each client has, ensuring the greater influence of the client with more data on the global model. This process is repeated for $\mathcal{T}$ times until the global model's loss function is minimal.

### 3.2.2 Algorithm

---

**Algorithm 1** Fed-GNODEFormer Algorithm

---

1: **Input:** T, t, C, $\theta_G^0$
2: Initialise $\theta_G^0$
3: **for** each round $R = 0, 1, 2, 3, ..., T-1$ **do**
4:     Sample Devices $\mathcal{S} \subseteq \mathcal{C}$
5:     **for** each client $s_i \in \mathcal{S}$ **in parallel do**
6:         $\theta_{s_i}^R \leftarrow UpdateClient(R, \theta_G^R)$
7:     **end for**
8:     $\theta_G^R \leftarrow \sum_{s_i=0}^{s_i=n} \frac{s_i}{\mathcal{S}} \theta_{s_i}^R$
9: **end for**
10:
11: **procedure** UPDATECLIENT(R, $\theta^R$)**:**
12:     $\mathcal{L}_{s_i} \leftarrow \text{NormalisedGraphLaplacian}(\mathcal{I}_n, \mathcal{A}, \mathcal{D})$
13:     $\gamma_{s_i}, \mathcal{U}_{s_i} \leftarrow \text{EigenDecomposition}(\mathcal{L}_{s_i})$
14:     **for** each round $r = 0, 1, 2, 3, \ldots, t-1$ **do**
15:         $\hat{y}_{s_i}, \gamma_{s_i}^{new} \leftarrow \text{GNODEFormer}(\gamma_{s_i}, \mathcal{U}_{s_i}, \beta_{s_i})$
16:         $\theta_{s_i}^r \leftarrow \underset{\theta_{s_i}}{\arg\min} \; \mathcal{L}_{CE}(\hat{y}_{s_i}[train_{s_i}], labels_{s_i}[train_{s_i}])$
17:     **end for**
18:     Return $\theta_{s_i}^R$
19: **end procedure**

---

## 4 Experiments

In this section, we conduct a wide variety of experiments on various heterophilic and homophilic graph datasets to verify the effectiveness of our approach. We compare our model against several state-of-the-art models including FedGCN (0 Hop, 1 Hop, 2 Hop) and FedSAGE, to demonstrate the improvements our model offers in different heterophilic and homophilic graph settings.

### 4.1 Dataset

For our experiments, we selected a diverse set of standard heterophilic and homophilic graph datasets to evaluate the robustness of our approach across different scenarios. The Cora and Citeseer datasets represent citation networks, and the Photo dataset is an Amazon co-purchase graph, all exhibiting homophilic properties. In contrast, the Chameleon and Squirrel datasets are heterophilic Wikipedia networks, while the Actor dataset corresponds to a film-related network. Each dataset includes nodes (representing distinct entities), edges (denoting relationships), node features, and class labels specific to their respective domains. Detailed information and summaries of these datasets are provided in the appendix.

### 4.2 Non independent and identically distributed Data

In FL-GNNs, non-IID data poses a significant challenge due to varying distributions across clients. We simulate these conditions using label Dirichlet partitioning, which controls distribution heterogeneity. The concentration parameter ($\alpha$) of the Dirichlet distribution determines data skewness: higher $\alpha$ values ($>= 1$) create near-IID settings, while lower values ($< 1$) result in highly skewed distributions, mimicking real-world non-IID scenarios.

In highly non-IID settings (low $\alpha$), local model updates can diverge significantly, leading to slower convergence, higher communication overhead, and potential model biases due to imbalance. A moderate $\alpha$ balances non-IID characteristics with learning efficiency, crucial for developing robust FL algorithms under realistic data heterogeneity.

Our proposed architecture is specifically designed to address these non-IID conditions in FL settings more effectively than existing methods, achieving better convergence and generalization performance.

### 4.3 Experimental setup

We conduct our experiments using a setup with 5 clients and a centralized method to evaluate our model's performance in both federated and centralized settings. To distribute data among the clients, we use a Dirichlet label distribution with a parameter $\alpha$ for all datasets. This approach allows us to assess the model's robustness and effectiveness under different data distribution scenarios, reflecting realistic federated learning environments.

In this setup, we compare the performance of the State-of-the-art Spectral Graph Transformer integrated with Neural Ordinary Differential Equations (GNODEFormer). We experiment with different ODE methods, specifically Runge-Kutta-2 (RK-2) and Runge-Kutta-4 (RK-4), across various datasets. We then compare our model's accuracy against existing state-of-the-art federated learning models, such as FedGCN with 0-Hop, 1-Hop, and 2-Hop configurations, and FedSAGE+ (Zhang et al., 2021), on both homophilic and heterophilic graph datasets.

### 4.4 Results

We present the performance evaluation of our proposed architecture, GNODEFormer, in both centralized and federated settings.

Table 1: Accuracy comparison of different models on heterophilic and homophilic datasets.

| Model | Heterophilic | | | Homophilic | | |
|---|---|---|---|---|---|---|
| | Chameleon | Squirrel | Actor | Cora | Citeseer | Photo |
| GCN | 59.61±2.21 | 46.78±0.87 | 33.23±1.16 | 87.14±1.01 | 79.86±0.67 | 88.26±0.73 |
| GAT | 63.13±1.93 | 44.49±0.88 | 33.93±2.47 | 88.03±0.79 | 80.52±0.71 | 90.94±0.68 |
| JacobiConv | 74.20±1.03 | 57.38±1.25 | 41.17±0.64 | 88.98±0.46 | 80.78±0.79 | 95.43±0.23 |
| Graphormer | 54.49±3.11 | 36.96±1.75 | 38.45±1.38 | 67.71±0.78 | 73.30±1.21 | 85.20±4.12 |
| Specformer | 74.72±1.29 | 64.64±0.81 | 41.93±1.04 | 88.57±1.01 | 81.49±0.94 | 95.48±0.32 |
| GNODEFormer (RK-2) | 79.04±0.29 | 66.26±0.26 | **45.15±0.10** | 88.30±0.05 | 83.85±0.15 | **95.94±0.02** |
| GNODEFormer (RK-4) | **79.41±0.02** | **66.65±0.03** | 44.56 ± 0.39 | **89.40±0.28** | **84.10±0.12** | 95.65 ± 0.02 |

Table 2: Accuracy results for Homophilic non-IID for 5 clients. Bold is the highest accuracy while underlined is the second highest.

| Client=5 | Homophilic Graphs | | | | | |
|---|---|---|---|---|---|---|
| Dataset | Cora | | Citeseer | | Photo | |
| $\alpha$ | 0.01 | 0.1 | 0.01 | 0.1 | 0.01 | 0.1 |
| FedGCN - 0 | 87.73% ± 0.31% | 81.87% ± 1.84% | 74.40% ± 0.49% | 70.03% ± 2.42% | 16.78% ± 10.43% | 34.53% ± 4.39% |
| FedGCN - 1 | 82.43% ± 0.54% | 81.67% ± 0.09% | 70.60% ± 0.92% | 70.10% ± 0.57% | 17.12% ± 15.98% | 36.08% ± 3.33% |
| FedGCN - 2 | 81.17% ± 0.33% | 81.03% ± 0.09% | 68.83% ± 1.23% | 69.47% ± 0.46% | 18.29% ± 14.28% | 12.89% ± 18.08% |
| FedSAGE+ | 87.83% ± 0.61% | 81.80% ± 1.61% | 74.73% ± 0.82% | 69.63% ± 1.89% | 16.19% ± 16.38% | 23.46% ± 16.88% |
| Fed-GNODEFormer(RK-2) | 89.28% ± 1.97% | 84.32% ± 0.23% | 74.54% ± 1.41% | 74.18% ± 1.91% | 93.50% ± 0.45% | 89.46% ± 10.27% |
| Fed-GNODEFormer(RK-4) | **89.64% ± 0.95%** | **85.16% ± 0.66%** | **75.18% ± 1.02%** | **74.22% ± 1.24%** | **96.76% ± 0.72%** | **94.94% ± 2.17%** |

Table 3: Accuracy results for Heterophilic non-IID for 5 clients. **Bold** is the highest accuracy while underlined is the second highest.

| Client=5 | Heterophilic Graphs | | | | | |
|---|---|---|---|---|---|---|
| Dataset | Chameleon | | Squirrel | | Actor | |
| $\alpha$ | 0.01 | 0.1 | 0.01 | 0.1 | 0.01 | 0.1 |
| FedGCN - 0 | 32.07% ± 10.36% | 27.70% ± 8.42% | 22.59% ± 4.11% | 22.59% ± 4.11% | 27.61% ± 2.03% | 27.35% ± 6.09% |
| FedGCN - 1 | 26.34% ± 7.25% | 20.12% ± 2.98% | 20.84% ± 1.89% | 20.84% ± 1.89% | 22.30% ± 3.53% | 23.68% ± 2.72% |
| FedGCN - 2 | 27.41% ± 7.96% | 22.06% ± 4.54% | 21.28% ± 2.12% | 21.28% ± 2.12% | 21.73% ± 3.77% | 23.47% ± 2.51% |
| FedSAGE+ | 32.76% ± 8.66% | 28.19% ± 8.70% | 22.59% ± 4.11% | 22.59% ± 4.11% | 27.61% ± 2.03% | 27.35% ± 6.09% |
| Fed-GNODEFormer(RK-2) | 68.45% ± 0.75% | 60.78% ± 0.32% | 53.32% ± 4.34% | 43.74% ± 1.60% | 41.54% ± 21.18% | **50.63% ± 23.36%** |
| Fed-GNODEFormer(RK-4) | **70.59% ± 1.99%** | **64.22% ± 3.21%** | **53.72% ± 5.55%** | **45.11% ± 0.41%** | **44.42% ± 20.91%** | 41.63% ± 18.01% |

On heterophilic datasets(Table 1), GNODEFormer achieves state-of-the-art results, outperforming existing models by notable margins on Chameleon (79.41%) and Squirrel (66.65%) datasets. For homophilic graphs, it shows competitive performance, achieving the highest accuracy on the Photo dataset (95.94%). In federated learning with extreme non-IID conditions ($\alpha = 0.01$), GNODEFormer consistently outperforms other approaches across both homophilic and heterophilic graphs. Its strength is further demonstrated by maintaining competitive performance across varying levels of data heterogeneity, as shown in 4.5. The model's success can be attributed to its innovative architecture combining neural ODEs with transformer(Vaswani, 2017) enabling more flexible modeling of graph dynamics. This design proves particularly effective in heterophilic and non-IID scenarios, with potential applications in social network analysis, recommendation systems, and fraud detection. This robustness in extreme non-IID settings is attributed to two core mechanisms in Fed-GNODEFormer's architecture: spectral graph methods and Neural ODEs. Spectral GNNs leverage the graph spectrum, enabling each client to learn global features rather than being restricted to local neighborhoods(Bo et al., 2023). This mitigates the impact of heterogeneous data distributions by learning patterns that generalize across clients. Neural ODEs further enhance this adaptability by dynamically adjusting the computation at each client based on its data complexity, allowing personalized yet aligned updates to the global model(Chen et al., 2018). Together, these mechanisms ensure Fed-GNODEFormer achieves better accuracy and convergence across clients, even under extreme non-IID conditions. However, GNODEFormer faces challenges in convergence speed and computational requirements, especially for larger heterophilic graphs, See Appendix D for scalability results. Future work should focus on improving efficiency and scalability while maintaining the model's strong performance across diverse graph types and data distributions.

### 4.5 Ablation

Table 3a shows the performance of our architecture across different $\alpha$ values. The model achieves its highest accuracy at $\alpha = 1.0$ (66.05%) and its lowest at $\alpha = 0.5$ (62.87%), with low standard deviation across all settings. These results demonstrate the robustness of our approach, maintaining stable performance even in non-IID scenarios.

To further understand the contribution of individual components to robustness, we conducted an ablation study (Table 4) by evaluating two variants: (i) Fed-GNODEFormer without the Neural-ODE Block, highlighting its role in adapting to non-IID data, and (ii) without the Residual Layer History, showing its impact on normalizing intermediate representations.

The results from these ablations underline the synergistic role of Neural-ODEs and Residual Layer History in achieving superior performance, particularly in challenging non-IID scenarios.

| $\alpha$ | Avg (%) | STD (%) |
|---|---|---|
| 0.1 | 64.71 | 0.53 |
| 0.2 | 64.32 | 1.25 |
| 0.3 | 63.74 | 0.39 |
| 0.4 | 64.67 | 0.35 |
| 0.5 | 62.87 | 0.20 |
| 0.6 | 64.58 | 0.28 |
| 0.7 | 64.00 | 0.77 |
| 0.8 | 64.23 | 0.22 |
| 0.9 | 65.21 | 0.54 |
| 1.0 | 66.05 | 0.20 |

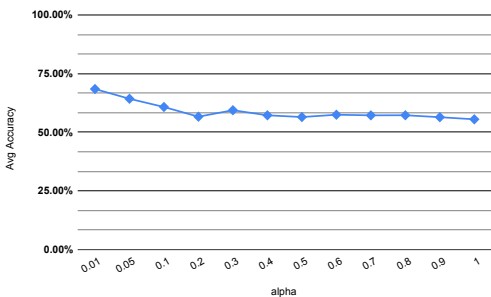

(a) Model Performance Across Different $\alpha$ Values.  (b) Model Performance on varying IID values.

Figure 3: Comparison of Model Performance on chameleon. The table (left) shows the average accuracy and standard deviation of the model across different $\alpha$ values for rk-2 model. The adjacent graph (right) provides a visual representation that further illustrates the trend in model performance, offering a clearer understanding of how the model adapts to varying data distributions.

Table 4: Component-wise ablation of Fed-GNODEFormer: We have compared the (1) and (2) with the original results of the Fed-GNODEFormer for the 3 varying non-iid $\alpha$ values.

| Methods | Homophilic | | | Heterophilic | | |
|---|---|---|---|---|---|---|
| Dataset | Citeseer | | | Squirrel | | |
| $\alpha$ | 0.01 | 0.05 | 0.1 | 0.01 | 0.05 | 0.1 |
| Fed-GNODEFormer Without the Neural ODE block | 73.56% | 73.1% | 71.2% | 43.22% | 42.26% | 40.44% |
| Fed-GNODEFormer (RK-2) without Residual Layer | 73.98% | 74.01% | 73.54% | 51.26% | 43.94% | 42.18% |
| Fed-GNODEFormer (RK-4) without Residual Layer | 74.85% | 74.36% | 74.18% | 52.13% | 44.37% | 43.67% |
| Fed-GNODEFormer (RK-2) | 74.54% | **74.58%** | 74.18% | 53.32% | **46.78%** | 43.74% |
| Fed-GNODEFormer (RK-4) | **75.18%** | 74.44% | **74.22%** | **53.72%** | 46.22% | **45.11%** |

The component-wise ablation results of Fed-GNODEFormer on CiteSeer (homophilic) and Squirrel (heterophilic) under different non-iid levels ($\alpha$) are shown in Table 4. Performance is severely reduced in all settings when the Neural ODE block is removed, underscoring its significance, especially on heterophilic graphs like Squirrel. The residual layer's function in stabilizing learning is suggested by the smaller but consistent negative impact when it is absent. RK-2 performs marginally better on heterophilic datasets, but RK-4 typically beats RK-2 on homophilic ones. Performance deteriorates as $\alpha$ increases, particularly for heterophilic graphs, highlighting the difficulties caused by high non-iidness.

## 5 Conclusion

In this work, we present GNODEformer, a novel federated learning method for Graph Neural Networks (GNNs) that effectively addresses the challenges of training on non-IID data in both homophilic and heterophilic graph settings. Our approach combines spectral GNNs with neural ordinary differential equations (ODEs) to enhance information capture and model adaptability across diverse graph types. The results show that GNODEformer significantly improves performance on non-IID heterophilic graphs and achieves competitive accuracy on homophilic graphs and IID data, outperforming several state-of-the-art methods. For future work, there are two promising directions. The first is developing a unified model that can seamlessly handle both IID and non-IID data, enhancing the flexibility and applicability of GNODEformer. The second is improving the runtime efficiency of federated learning setups, especially in large-scale and resource-constrained environments, to further optimize our approach.

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

# A   Appendix

## A.1   Datasets

| Dataset | Type | Nodes | Edges | Features | Classes |
|---------|------|-------|-------|----------|---------|
| Cora | Homophily | 2,708 | 5,429 | 1,433 | 7 |
| Citeseer | Homophily | 3,327 | 4,732 | 3,703 | 6 |
| Photo | Homophily | 7,650 | 11,081 | 745 | 8 |
| Chameleon | Heterophily | 2,277 | 36,101 | 2,325 | 5 |
| Squirrel | Heterophily | 5,201 | 217,073 | 2,089 | 5 |
| Actor | Heterophily | 7,600 | 33,544 | 932 | 5 |

Table 5: Summary of Datasets used for node classification

1. **Cora**: This is a citation network graph dataset often used in graph machine learning tasks, known for its homophilic properties. In this dataset, nodes represent documents, and edges represent citation links between them. Each node comes with a sparse feature vector, representing the keywords in the document, and a class label.

2. **Citeseer**: Similar to cora, this dataset is also a homophilic citation network dataset with a challenging classification task due to its sparse connectivity.

3. **Photo**: Amazon Photo is a homophilic dataset and consists segments of the Amazon co-purchase graph (McAuley et al., 2015), where nodes represent goods, edges indicate that two goods are frequently bought together, node features are bag-of-words encoded product reviews, and class labels are given by the product category.

4. **Chameleon**: This dataset consists of nodes that represent articles from the English Wikipedia, with edges indicating mutual links between these articles. Each node has features based on the presence of specific nouns in the article. The nodes are categorized into five classes according to their average monthly traffic. Due to its low level of homophily, the dataset poses a more challenging classification problem (Rozemberczki et al., 2021).

5. **Squirrel**: Similar to Chameleon, this dataset is heterophilic and is part of a collection of datasets representing page-page networks on specific topics such as chameleons, crocodiles, and squirrels. In these datasets, nodes correspond to Wikipedia articles, and edges indicate mutual links between them. Node features are derived from informative nouns appearing in the article text, while target labels represent the average monthly traffic for each article from October 2017 to November 2018. The datasets include detailed statistics on the number of nodes and edges (Rozemberczki et al., 2021).

6. **Actor**: This dataset represents the subgraph that includes only actors from a larger film-director-actor-writer network (Tang et al., 2009). Each node represents an actor, and an edge between two nodes indicates that the actors appear together on the same Wikipedia page. The features of each node are derived from specific keywords found on those Wikipedia pages. The nodes are categorized into five groups based on the words in the actors' Wikipedia entries.

## A.2   Discussion on Results from Table 5 ($\alpha = 0.05$ across graphs)

In homophilic graphs (Cora, CiteSeer, and Photo datasets), GNODEFormer (RK-2 and RK-4) consistently outperforms the baseline models (FedGCN-0, FedGCN-1, FedGCN-2, and FedSage+). Notably, on the Cora dataset, GNODEFormer (RK-2) achieves the highest accuracy of 87.52%, surpassing the next-best model (FedSage+) by a margin of 2.25%. A similar trend is observed on the CiteSeer and Photo datasets, where GNODEFormer (RK-2) and GNODEFormer (RK-4) provide the top performances, with GNODEFormer

(RK-4) achieving the best accuracy of 88.69% on the Photo dataset. These results reinforce the adaptability of GNODEFormer to homophilic structures, demonstrating its ability to leverage node similarity more effectively than the traditional federated GCN and GAT models.

For heterophilic graphs (Chameleon, Squirrel, and Actor datasets), GNODEFormer shows a particularly strong performance, especially in the RK-4 setting. On the Chameleon dataset, GNODEFormer (RK-4) achieves 65.03%, outperforming all other models, including FedSage+. Similarly, in the Squirrel and Actor datasets, GNODEFormer (RK-4) yields the highest accuracies of 46.22% and 51.53%, respectively, showcasing its superior ability to manage heterophilic relationships, where the node connections are more complex and less homogeneous. However, as noted earlier, the model's improved performance comes at the cost of increased computational requirements, particularly in larger and more complex heterophilic graphs, which may slow down convergence in real-world federated learning scenarios. While the model excels in accuracy, especially in the RK-4 configuration, future work could focus on optimizing the computational efficiency to ensure faster convergence, particularly for larger, heterophilic datasets.

We also highlight a very insteresting observation. In extreme non-IID settings, since the data distribution across clients is very uneven. For example, users with similar interests or demographics could be clustered within a particular country, leading to an imbalanced distribution of node classes across different clients (Yao et al., 2023). In such situations, aggregating information over many hops could lead to the model primarily learning from nodes with similar labels within each client. This can lead to node embeddings that are too similar, ultimately hindering the model's ability to discriminate between different node classes, which is the essence of oversmoothing (Rusch et al., 2023). This leads to FedGCN(2-hop) should perform worse than FedGCN(0-hop) in extreme non-IID settings.

Table 6: Results for $\alpha$=0.05 for across graphs

| Client=5 | Homophillic | | | Heterophillic | | |
|---|---|---|---|---|---|---|
| Dataset | Cora | citeseer | photo | chameleon | squirrel | actor |
| $\alpha$ | 0.05 | 0.05 | 0.05 | 0.05 | 0.05 | 0.05 |
| FedGCN - 0 | 85.27% ± 2.17% | 74.37% ± 0.46% | 24.97% ± 11.61% | 27.05% ± 5.01% | 22.59% ± 4.11% | nan ± nan |
| FedGCN - 1 | 82.10% ± 0.64% | 70.70% ± 0.57% | 33.79% ± 3.30% | 25.56% ± 6.34% | 20.84% ± 1.89% | nan ± nan |
| FedGCN - 2 | 80.83% ± 0.81% | 68.43% ± 0.66% | 33.76% ± 3.33% | 22.71% ± 2.27% | 21.28% ± 2.12% | nan ± nan |
| FedSage+ | 85.27% ± 2.17% | 74.37% ± 0.46% | 33.84% ± 3.41% | 27.05% ± 5.01% | 22.59% ± 4.11% | nan ± nan |
| NODEphormer(RK-2) | **87.52% ± 1.97%** | **74.58% ± 0.80%** | **93.70% ± 2.86%** | 64.29% ± 2.44% | **46.78% ± 2.95%** | 44.04% ± 17.30% |
| NODEphormer(RK-4) | 87.30% ± 2.28% | 74.44% ± 1.02% | 88.69% ± 0.49% | **65.03% ± 5.06%** | 46.22% ± 4.85% | **51.53% ± 25.28%** |

## A.3 Cost and Communication analysis

We provide cost and communication analysis of our proposed architecture in FL setup. In communication analysis table we provide statistic for parameters and memory usage of transferred model in bytes.

Table 7: Communication analysis of GNODEFormer

| Dataset | Parameters | Size in B |
|---|---|---|
| cora | 140218 | 560872 |
| citeseer | 285426 | 1141704 |
| photo | 96258 | 385032 |
| chameleon | 197162 | 788648 |
| squirrel | 182058 | 728232 |
| actor | 108010 | 432040 |

Table 8 provides an analysis of the time required for Fed-GNODEFormer to complete an entire experiment as well as the time taken for one local epoch across various datasets, comparing the second-order Runge-Kutta and fourth-order Runge-Kutta methods.

It is evident that RK-4, though delivering superior performance in terms of accuracy (as discussed in previous sections), comes at the cost of increased computational time. Across all datasets, the complete experiment

time for RK-4 is significantly longer compared to RK-2. For instance, on the Photo dataset, RK-4 takes 392.61 seconds, which is approximately 1.65x longer than RK-2, which completes in 237.371 seconds. Similarly, the Squirrel dataset shows a similar trend, where RK-4 requires 245.274 seconds, while RK-2 only takes 187.184 seconds.

This increased computational time with RK-4 is also reflected in the time per local epoch. For all datasets, the time taken per epoch is consistently higher for RK-4. For example, on the Actor dataset, RK-4 takes 3.946 seconds per epoch, which is significantly higher than the 2.361 seconds required for RK-2. Similarly, on the Photo dataset, RK-4 takes nearly 4 seconds (3.919 seconds), which is again much longer than the 2.368 seconds for RK-2.

From these results, it is clear that while RK-4 provides better model performance in terms of accuracy, especially on complex datasets like heterophilic graphs (Chameleon, Squirrel, Actor), it requires more computational time, both at the level of a single epoch and the overall training process. The increased complexity of RK-4, particularly its ability to better capture the complex relationships in heterophilic datasets, seems to directly translate to increased resource demands.

This trade-off between performance and computation must be carefully considered in real-world scenarios, especially where computational resources are limited or when working with larger datasets. In such cases, RK-2 may be preferred as it offers a balance between reasonable accuracy and faster computation times, especially in homophilic graphs like Cora and CiteSeer. However, for tasks where accuracy and model performance are critical, and computational resources are less of a concern, RK-4 remains a better choice, despite its slower convergence.

Table 8: Time Analysis of GNODEFormer

| Dataset | Method | Complete Experiment (Seconds) | One local epoch (Seconds) |
|---|---|---|---|
| Cora | RK-2 | 129.715 | 1.293 |
| | RK-4 | 149.631 | 1.492 |
| Citeseer | RK-2 | 133.908 | 1.335 |
| | RK-4 | 175.14 | 1.747 |
| Photo | RK-2 | 237.371 | 2.368 |
| | RK-4 | 392.61 | 3.919 |
| Chameleon | RK-2 | 128.997 | 1.286 |
| | RK-4 | 144.147 | 1.437 |
| Squirrel | RK-2 | 187.184 | 1.867 |
| | RK-4 | 245.274 | 2.447 |
| Actor | RK-2 | 263.627 | 2.361 |
| | RK-4 | 395.259 | 3.946 |

### A.4 Compute Resources

We utilized an NVIDIA GeForce RTX 4090 with 24 GB of memory and NVIDIA A100 GPUs configured with MIG instances, each equipped with 20 GB of memory, for our Federated Learning (FL) setup. All datasets were trained in a federated setting using the NVIDIA GeForce RTX 4090 and A100. For centralized testing, the NVIDIA A100 was used for the Cora, Citeseer, Chameleon, and Squirrel datasets, while the NVIDIA A40 with 48 GB of memory was employed for training the Photo and Actor datasets using the Runge-Kutta 4th-order (RK-4) method. For cost and communication analysis, a single NVIDIA A100 GPU operating in MIG mode with 20 GB of memory was utilized.

### A.5 Comparing GNODEFormer across graph types

In Figure 4, we plot accuracy vs. epochs to show convergence of our centralised model on both type of graphs.

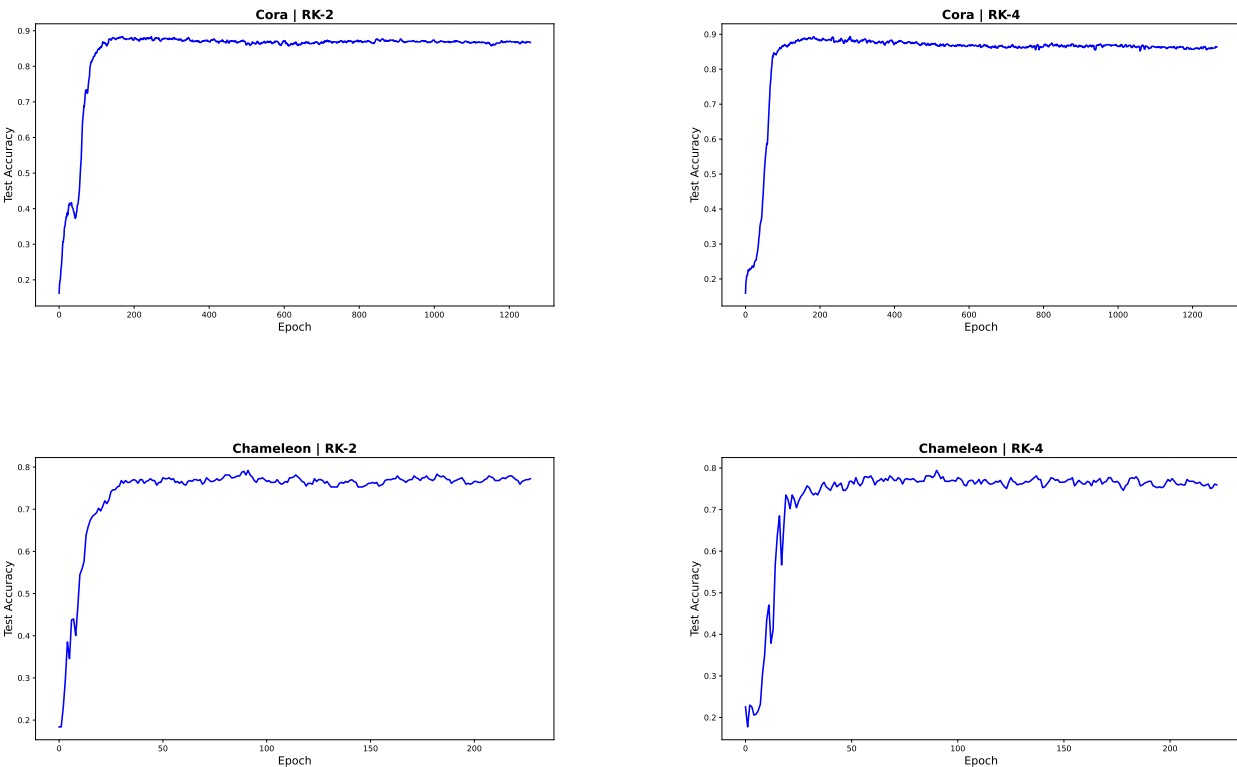

Figure 4: Comparative performance of Cora(Homophilic) and Chameleon(Heterophilic) datasets using RK-2 and RK-4 methods.

## B   More Ablation

### B.1   Visualizations and Plots for Learned Spectral Filters in GNODEFormer

We see the filter learned by the our model for homophillic graphs are low-pass filters and are better than Specformer. Same is true for band-pass filters for heterophillic graphs too.

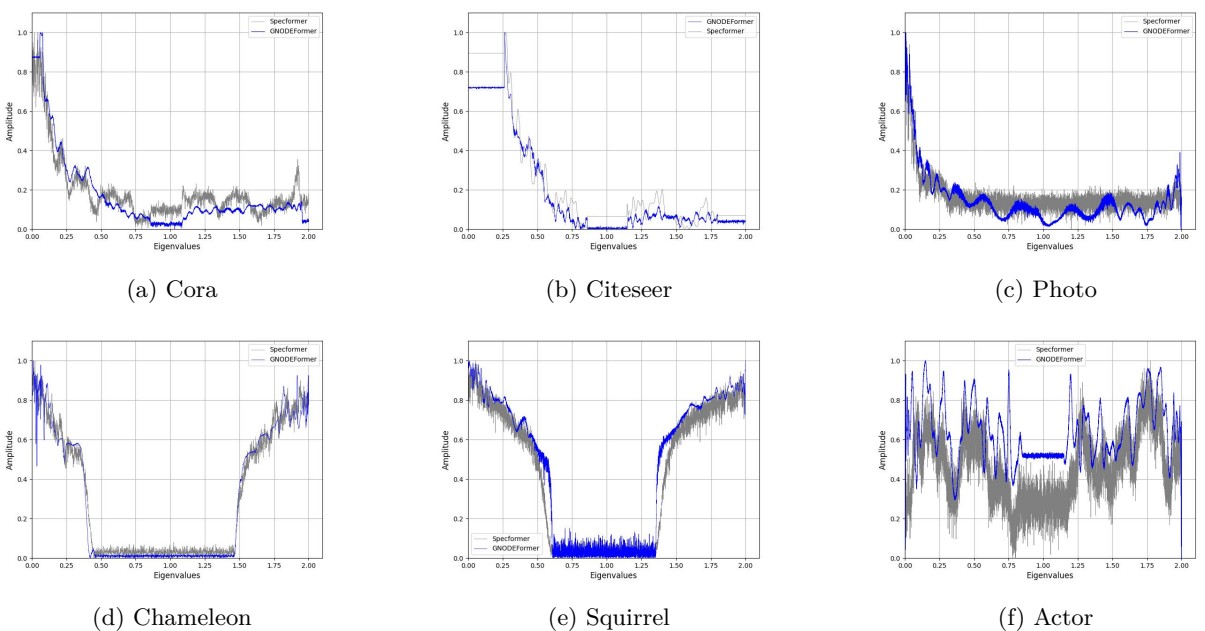

Figure 5: Eigenvalue plots for different datasets.

## C   Mathematical Representation of ODE Integration in Transformer Blocks

Here we provide a mathematical foundation for integrating Ordinary Differential Equations (ODEs) into Transformer architectures.

### C.1   Residual Block as an Euler Discretization

A standard residual block in a Transformer can be expressed as:

$$y_{t+1} = y_t + F(y_t, \theta_t),$$

where $y_t$ is the input at step $t$, $\theta_t$ represents block parameters, and $F(y_t, \theta_t)$ denotes the function computed by the block (e.g., self-attention or feedforward network).

This formulation aligns with the Euler method for solving ODEs:

$$y(t + \Delta t) = y(t) + \Delta t \cdot F(y(t), \theta(t)),$$

where $\Delta t$ is the step size (set to 1 in standard Transformers). As $\Delta t \to 0$, this discretization leads to the continuous ODE:

$$\frac{dy(t)}{dt} = F(y(t), \theta(t)).$$

Thus, a residual block corresponds to a single step in solving this ODE.

## C.2 Higher-Order Runge-Kutta Methods

To improve accuracy, higher-order ODE solvers, such as the Runge-Kutta (RK) methods, are proposed for designing Transformer blocks. The general form of an explicit $n$-step RK method is:

$$y_{t+1} = y_t + \sum_{i=1}^{n} \gamma_i F_i,$$

where $F_i$ are intermediate approximations computed iteratively as:

$$F_i = F\left(y_t + \sum_{j=1}^{i-1} \beta_{ij} F_j, \theta_t\right),$$

and $\gamma_i, \beta_{ij}$ are method-specific coefficients.

This equation can be directly mapped to a neural network's layer-based representation:

$$z^{l+1} = z^l + \sum_{i=1}^{p} w_i f_\theta\left(z^l + \sum_{j=1}^{i-1} a_{ij} k_j\right),$$

where:

- $z^{(l)}$ is the input to the layer (analogous to $y_t$),

- $z^{(l+1)}$ is the output of the layer (analogous to $y_{t+1}$),

- $p$ is the order of the Runge-Kutta method (e.g., 2, 3, or 4),

- $f_\theta$ represents the neural network layer function with parameters $\theta$,

- $w_i$ corresponds to $\gamma_i$, serving as weights for combining each intermediate $k_i$,

- $a_{ij}$ corresponds to $\beta_{ij}$, defining the coefficients for intermediate steps,

- $k_i$ are intermediate values representing estimates of changes over a step.

In this formulation:

- Each layer corresponds to a temporal evolution step in solving the ODE.

- The weights $\gamma_i$ in the RK method are represented by $w_i$ in the network.

- The coefficients $\beta_{ij}$ map to $a_{ij}$, controlling intermediate computations.

- The function $F(y_t, \theta_t)$ in the RK method is modeled as $f_\theta$, parameterized by the network.

This parallel demonstrates how higher-order RK methods naturally extend to neural networks, enabling precise approximations of ODE solutions while maintaining interpretability in terms of temporal dynamics.

We have used popular variants like RK2 (second-order) and RK4 (fourth-order) as they offer more precise approximations of ODE solutions within each Transformer block.

## C.3 Coefficient Learning

Learning coefficients $\gamma_i$ (or weights $w_i$ in the network) during training further optimizes model performance by adaptively weighting intermediate approximations. This enhances the expressiveness and efficiency of the Transformer architecture.

### C.4 Working of Runge-Kutta variants

Based on the equations below -

$$z^{l+1} = z^l + \sum_{i=1}^{p} w_i f_\theta \left( x_n + \sum_{j=1}^{i-1} a_{ij} k_j \right) \tag{7}$$

where $z^{(l)}$ is the input to the layer, $z^{(l+1)}$ is the output of the layer, $p$ is the order of the Runge-Kutta method (2, 3, or 4), $f_\theta$ represents the neural network layer function with parameters $\theta$, $w_i$ are the weights for combining each $k_i$ term, $a_{ij}$ are the coefficients for the intermediate steps, $k_i$ are intermediate values representing estimates of the change in $x$ over a step, and $k_j$ are the previously calculated $k$ values used in computing the current $k_i$.

We show the calculation of RK2 and RK4 methods by putting relevant values.

#### C.4.1 Working of RK2 Method

$$w = \begin{bmatrix} \frac{1}{2} & \frac{1}{2} \end{bmatrix} \tag{8}$$

$$a = \begin{bmatrix} 0 & \\ 1 & 0 \end{bmatrix} \tag{9}$$

Expanded form:

$$k_1 = f_\theta(z^l) \tag{10}$$
$$k_2 = f_\theta(z^+ k_1) \tag{11}$$

The final step calculation:

$$z^{(l+1)} = z^l + \frac{1}{2} k_1 + \frac{1}{2} k_2 \tag{12}$$

#### C.4.2 Working of RK4 Method

$$w = \begin{bmatrix} \frac{1}{6} & \frac{1}{3} & \frac{1}{3} & \frac{1}{6} \end{bmatrix} \tag{13}$$

$$a = \begin{bmatrix} 0 & & & \\ \frac{1}{2} & 0 & & \\ 0 & \frac{1}{2} & 0 & \\ 0 & 0 & 1 & 0 \end{bmatrix} \tag{14}$$

Expanded form:

$$k_1 = f_\theta(z^l) \tag{15}$$
$$k_2 = f_\theta(z^l + 0.5 k_1) \tag{16}$$
$$k_3 = f_\theta(z^l + 0.5 k_2) \tag{17}$$
$$k_4 = f_\theta(z^l + k_3) \tag{18}$$

$$z^{(l+1)} = z^l + \frac{1}{6} k_1 + \frac{1}{3} k_2 + \frac{1}{3} k_3 + \frac{1}{6} k_4 \tag{19}$$

# D   Scalability with Graph Size

Our method inherits standard scaling challenges of GNNs, such as memory and computation overhead for large graphs. Existing solutions, like sampling techniques or graph sparsification, can be adapted to address these issues and explored in future work. Section A.3 provides a detailed analysis of the computational costs and communication overhead in a federated learning (FL) setup, comparing second-order (RK-2) and fourth-order (RK-4) Runge-Kutta solvers.

Table 8 shows that RK-4 delivers superior accuracy on complex heterophilic graphs but at a higher computational cost (e.g., $1.65\times$ longer on the Photo dataset). RK-2, while faster, is effective for homophilic graphs or resource-constrained scenarios. This trade-off demonstrates that our method scales effectively, balancing computational demands and performance based on application requirements.

We also conducted extensive experiments with GNODEFormer on large-scale homophilic and heterophilic graphs. Specifically, we tested on Penn94 (heterophilic) (Lim et al., 2021), which contains 41,554 nodes and 1,362,229 edges, and ogbn-arXiv (homophilic) (Lim et al., 2021), which has 169,343 nodes and 1,166,243 edges. Below are the experimental results:

| Model | Penn94 | OGBN-arXiv |
|---|---|---|
| GCN | $82.47 \pm 0.27$ | $71.74 \pm 0.29$ |
| GAT | $81.53 \pm 0.55$ | $71.82 \pm 0.23$ |
| JacobiConv | $83.35 \pm 0.11$ | $72.14 \pm 0.17$ |
| Graphormer | OOM | OOM |
| SpecFormer | $84.32 \pm 0.32$ | $72.37 \pm 0.18$ |
| GNODEFormer (RK-2) | $84.96 \pm 0.06$ | $72.76 \pm 0.17$ |
| GNODEFormer (RK-4) | $84.98 \pm 0.04$ | $72.97 \pm 0.30$ |

Table 9: Performance comparison on large-scale graphs. OOM: Out of Memory.

Additionally, we analyzed the filters learned by GNODEFormer and compared them with those of SpecFormer at the 25th epoch, the 250th epoch, and the final epoch for both the Arxiv and Penn94 datasets. Given the large scale of these graphs, this intermediate analysis was essential for evaluating how our model learns over time and for understanding the evolution of the learned filters.

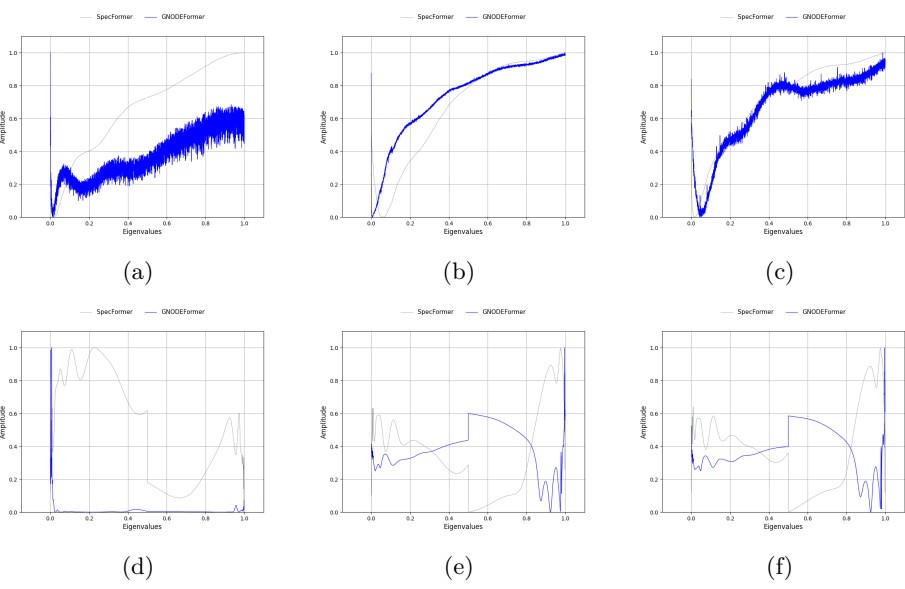

Figure 6: Learned filters at the 25th, 250th, and final epochs for Arxiv (a–c, top) and Penn94 (d–f, bottom).

# E    Hyperparameter Tuning Strategy

We followed standard hyperparameter tuning practices to ensure fair and consistent evaluation across all baselines and variants of our proposed model. Specifically, we performed grid search over the following parameters:

- Learning rates: $\{1e{-}4, 5e{-}4, 1e{-}3, 5e{-}3\}$

- ODE solver steps: $\{2, 4\}$

In addition, we made slight manual adjustments based on empirical observations during inference. For example, we used a slightly lower learning rate in cases with extremely low $\alpha$ values (i.e., high non-IID data settings) and significant heterophily, as this improved stability and convergence.

We also conducted an ablation study to evaluate the impact of key hyperparameters, including the ODE solver step size and model depth. These results are summarized in Appendix 4.5.

We also observed that model performance exhibits nuanced behavior across different $\alpha$ values. In general, higher $\alpha$ (i.e., more IID data) tends to improve average accuracy, though not uniformly across all models.

