# OpenReview forum: "Federated Spectral Graph Transformers Meet Neural Ordinary Differential Equations for Non-IID Graphs"
_TMLR — Accepted by TMLR_

### Review · Reviewer_tZrz · 2024-12-13

**Summary Of Contributions:**

The paper introduces a novel federated learning approach for Graph Neural Networks (GNNs) that combines spectral GNNs with neural ordinary differential equations (ODEs) to address non-IID data challenges in both homophilic and heterophilic graph settings.

**Audience:**

Yes

**Claims And Evidence:**

Yes

**Requested Changes:**

See Weaknesses.

**Strengths And Weaknesses:**

Strengths:
1. The paper provides a clear motivation for the proposed method, emphasizing the practical need for privacy-preserving graph learning.
2. The experimental results are comprehensive, demonstrating the method's effectiveness across a variety of graph datasets and non-IID conditions.

Weaknesses:
1. How does the proposed method scale with the size of the graph, especially when dealing with very large and complex graphs?
2. Are there any specific hyperparameter tuning strategies mentioned for the neural ODE components that are crucial for the model's performance?
3. There are several relevant methods missing in the manuscript that need to be discussed and analyzed: A Survey of Graph Neural Networks in Real world: Imbalance, Noise, Privacy and OOD Challenges.
4. What are the computational costs associated with the higher-order Runge-Kutta methods compared to simpler ODE solvers, and does this impact the practicality of the approach?

---

> ### Author Response · Authors · 2025-02-09
> **Response to Reviewer tZrz**
>
> We thank the reviewer for their comments on our work. We provide a detailed rebuttal for the same below.
> > How does the proposed method scale with the size of the graph, especially when dealing with very large and complex graphs?
>
> Our method inherits standard scaling challenges of GNNs, such as memory and computation overhead for large graphs. Existing solutions, like sampling techniques or graph sparsification, can be adapted to address these issues and explored in future work. Section A.3 provides a detailed analysis of the computational costs and communication overhead in an FL setup, comparing second-order (RK-2) and fourth-order (RK-4) Runge-Kutta solvers. Table 8 shows that RK-4 delivers superior accuracy on complex heterophilic graphs but at a higher computational cost (e.g., 1.65x longer on the Photo dataset). RK-2, while faster, is effective for homophilic graphs or resource-constrained scenarios. This trade-off demonstrates that our method scales effectively, balancing computational demands and performance based on application requirements.
>
> We conducted extensive experiments with GNODEFormer on large-scale homophilic and heterophilic graphs. Specifically, we tested on Penn94 (heterophilic) [1], which contains 41,554 nodes and 1,362,229 edges, and ogbn-arXiv (homophilic) [1], which has 169,343 nodes and 1,166,243 edges. Below are the experimental results:
>
> | Model               | Penn94          | OBGN-arXiv       |
> |---------------------|-----------------|------------------|
> | GCN                | 82.47 ± 0.27   | 71.74 ± 0.29    |
> | GAT                | 81.53 ± 0.55   | 71.82 ± 0.23    |
> | JacobiConv         | 83.35 ± 0.11   | 72.14 ± 0.17    |
> | Graphormer         | OOM            | OOM             |
> | SpecFormer         | 84.32 ± 0.32   | 72.37 ± 0.18    |
> | GNODEFormer (RK-2) | 84.9562 ± 0.06 | 72.762 ± 0.176  |
> | GNODEFormer (RK-4) | 84.9781 ± 0.04 | 72.973 ± 0.299  |
>
> Additionally, we analyzed the filters learned by GNODEFormer and compared them with those of SpecFormer at the 25th epoch, 250th epoch, and in the final filter plots. Given the large scale of these graphs, this intermediate analysis was essential to evaluate how our model learns over time and to understand the evolution of the learned filter plots. These will be added to the appendix of the revised version for further clarity and analysis.
>
> [1] Lim, Derek, et al. "Large scale learning on non-homophilous graphs: New benchmarks and strong simple methods." Advances in Neural Information Processing Systems 34 (2021): 20887-20902.
>
> > Are there any specific hyperparameter tuning strategies mentioned for the neural ODE components that are crucial for the model's performance?
>
> For the neural ODE components, we used standard hyperparameter tuning strategies, including grid search over learning rates, hidden dimensions, and the number of ODE solver steps. A careful ablation study was conducted, as highlighted in response to Reviewer 87kj, to understand the impact of these choices. We will include these details in the manuscript and provide further information in the appendix.
>
> > There are several relevant methods missing in the manuscript that need to be discussed and analyzed: A Survey of Graph Neural Networks in Real world: Imbalance, Noise, Privacy and OOD Challenges.
>
> We appreciate the reviewer highlighting the need to discuss additional relevant methods. However, our manuscript specifically targets the privacy challenge in training GNNs, aligning with the scope of federated learning and its application to non-IID graph data. While we acknowledge the broader challenges mentioned in the survey, incorporating an exhaustive analysis of these unrelated challenges (imbalance, noise, OOD) would detract from our focus. We thank the reviewer for pointing out this survey and will cite it to acknowledge its contributions in our paper.
>
> > What are the computational costs associated with the higher-order Runge-Kutta methods compared to simpler ODE solvers, and does this impact the practicality of the approach?
>
> Section A.3 discussed cost and communication analysis. Simpler ODE solvers, although faster than RK methods, do not deliver good accuracy. We ran tests on other Higher-order methods, such as the Dormand-pice method. We found it not suitable for our method as it is very expensive.

---

### Review · Reviewer_AfWR · 2024-12-23

**Summary Of Contributions:**

This paper explores Fed-GNODEFormer, a GNN model designed for FL that addresses the challenge of heterogeneous graph data distribution across clients. The model leverages spectral decomposition and ODEs to capture both structural and dynamic graph features, enhancing performance on complex and diverse graph datasets. The authors present numerical experiments to validate their methods.

**Audience:**

Yes

**Claims And Evidence:**

Yes

**Requested Changes:**

* Regarding the aforementioned W1, I recommend that the authors include a discussion on client assignment and consider conducting additional experiments with a larger number of clients, if needed.
* Regarding W2, I suggest the authors provide clearer explanations or revisions to the figures to enhance clarity and avoid potential confusion.
* Some citation styles in the paper need to be corrected. The authors should clarify the use of \citep and \citet.

**Strengths And Weaknesses:**

**Strengths**:
1. The paper is well-organized, with each component of the proposed method clearly explained and seems easy to follow.
2. The existing numerical results seems good and demonstrate the effectiveness of the proposed method.

**Weaknesses**:
1. I’m a little bit unclear on how client selection is conducted in your experiments, and the current setup with 5 clients seems somewhat limited.
2. Figure 2 is a bit confusing to me. I understand that GNODEFormer is designed for undirected graphs, but the figure depicts an example of a directed graph. Additionally, the proposed Fed-GNODEFormer appears to assume different graph structures across clients by “partitioning the global graph G into m subgraphs.” However, this partitioning process isn’t clearly illustrated in Figure 2.
3. In my view, the key factor driving Fed-GNODEFormer’s performance improvement in FL scenarios primarily stems from the advantages of the GNODEFormer architecture itself. This includes its strong performance under highly non-IID conditions, suggesting that much of the benefit can be attributed to GNODEFormer rather than the adjustment for federated learning.

---

> ### Author Response · Authors · 2025-02-09
> **Response to Reviewer AfWR**
>
> We thank the reviewer for their comments on our work. We provide a detailed rebuttal for the same below.
>
> >I’m a little bit unclear on how client selection is conducted in your experiments, and the current setup with 5 clients seems somewhat limited.
>
> We appreciate the reviewer’s question regarding client selection and the number of clients in our experiments. In our setup, we use random client selection, a widely accepted approach [1, 2, 3] in federated learning, as it replicates real-world scenarios and ensures fair representation without introducing biases [4] from specific selection heuristics. The choice to use five clients was motivated by practical considerations [1, 2, 3], balancing computational efficiency and the ability to capture meaningful insights, particularly in non-IID scenarios. This setup allows for comprehensive experimentation across various datasets within reasonable computational limits.
>
> [1] Gouissem, Ala, Zina Chkirbene, and Ridha Hamila. "A comprehensive survey on client selections in federated learning." Innovation and Technological Advances for Sustainability (2024): 417-428.
>
> [2] Fu, Lei, et al. "Client selection in federated learning: Principles, challenges, and opportunities." IEEE Internet of Things Journal (2023).
>
> [3] Rai S, Kumari A, Prasad DK. Client Selection in Federated Learning under Imperfections in Environment. AI. 2022; 3(1):124-145.
>
> [4] Cho, Yae Jee, Jianyu Wang, and Gauri Joshi. "Towards understanding biased client selection in federated learning." International Conference on Artificial Intelligence and Statistics. PMLR, 2022.
>
> > Figure 2 is a bit confusing to me. I understand that GNODEFormer is designed for undirected graphs, but the figure depicts an example of a directed graph. Additionally, the proposed Fed-GNODEFormer appears to assume different graph structures across clients by “partitioning the global graph G into m subgraphs.” However, this partitioning process isn’t clearly illustrated in Figure 2.
>
> We appreciate the reviewer’s feedback regarding Figure 2 and the potential for confusion. However, the directions and arrows are just used to show the working of the FL. It does not imply any specific directions to be followed or a directed graph. We will replace the directed graph example in Figure 2 with an undirected graph example to better align with Fed-GNODEFormer’s design. Additionally, we will revise the figure and the caption to clearly illustrate the partitioning process of the global graph G into m subgraphs, ensuring that the decentralized structure across clients is visually explicit.
>
> > In my view, the key factor driving Fed-GNODEFormer’s performance improvement in FL scenarios primarily stems from the advantages of the GNODEFormer architecture itself. This includes its strong performance under highly non-IID conditions, suggesting that much of the benefit can be attributed to GNODEFormer rather than the adjustment for federated learning.
>
> We thank the reviewer for recognizing this core strength of our work. Indeed, the strong performance of Fed-GNODEFormer in FL scenarios can be attributed to the strengths of the GNODEFormer architecture, including its robustness under highly non-IID conditions. However, as noted in our paper, our method was specifically designed with the federated learning setting in mind. We later observed that its design also extends well to centralized settings, as mentioned in our paper in Introduction section: "Notably, while our approach is tailored for federated learning, its design is versatile enough to perform well even in centralized settings, showcasing its broad applicability to various graph learning scenarios." This highlights that the architecture’s adaptability across both settings underscores its practical utility and relevance to broader graph learning challenges.
>
> > Some citation styles in the paper need to be corrected. The authors should clarify the use of \citep and \citet.
>
> We thank the reviewer for pointing out the inconsistency in citation styles. We will thoroughly review the manuscript to ensure the correct usage of \citep and \citet throughout the paper, and we will address this issue in the revised version.

---

### Review · Reviewer_87kj · 2025-01-27

**Summary Of Contributions:**

This paper proposes using the Runge-Kutta method to simulate the process of graph signal propagation in the frequency domain. The paper further extends the proposed GNN to the federated learning setting. The paper provides experiments on both centralized and federated learning settings to demonstrate that the proposed GNN has a better capacity to express the graphs (node classification) on various datasets under homogeneous and heterogeneous data distributions.

**Audience:**

Yes

**Claims And Evidence:**

No

**Requested Changes:**

Please address the weaknesses above.

Minor:
1. replace "alpha" with $\alpha$.
2. Correctly using \citet{} and \citep{}
3. The notations are confusing. $n$ is used as both the number of nodes, the number of layers, and the number of clients. $i,j$ are used as node index, steps in RK method, client index. $s_i$ and $c_i$ are used to denote client, etc. The paper is hard to understand.

**Strengths And Weaknesses:**

Strength:
1. The paper provides a novel idea of using high-order ODE solvers to simulate the message-passing processing in the GNN.
2. Numerical results on the datasets demonstrate strong improvement compared with existing methods.

Weakness:
1. Missing discussion on related works. Specifically, the author should provide more discussion on the connection between the proposed method and other GNN+ODE/PDE works (e.g., [R1], [R2]).
2. Unclear FL setting. FL+Graph data has many different settings, including graph/subgraph/edge/node/feature distributed settings. The paper should clarify the setting in the abstract and in the introduction.
3. It is counter-intuitive that more iid data (larger $\alpha$) results in worse performance than non-iid data. Although the authors claim it is due to a lack of sufficient parameter tuning, it is not convincing. More careful experiments are desired.

[R1] Zhuang J, Dvornek N, Li X, Duncan JS. Ordinary differential equations on graph networks.
[R2] Eliasof M, Haber E, Treister E. Pde-gcn: Novel architectures for graph neural networks motivated by partial differential equations. Advances in neural information processing systems. 2021 Dec 6;34:3836-49.

---

> ### Author Response · Authors · 2025-02-09
> **Response to Reviewer 87kj**
>
> We thank the reviewer for their comments on our work. We provide a detailed rebuttal for the same below.
>
> > Missing discussion on related works. Specifically, the author should provide more discussion on the connection between the proposed method and other GNN+ODE/PDE works (e.g., [R1], [R2]).
>
> Thank you for pointing this out. We will expand the discussion on related works in the revised version. Specifically, we will highlight how our method relates to and differentiates itself from other GNN+ODE/PDE approaches like [R1] and [R2].
>
> > Unclear FL setting. FL+Graph data has many different settings, including graph/subgraph/edge/node/feature distributed settings. The paper should clarify the setting in the abstract and in the introduction.
>
> We appreciate the reviewer’s feedback regarding the need for clarity on the federated learning (FL) setting used in our work. In Section 3.2.1, we describe our specific FL setup, which involves distributing graph data at the subgraph level across clients to simulate decentralized learning scenarios. We will revise the manuscript to include this clarification in the final draft in the abstract and introduction, ensuring better alignment with the broader FL+Graph literature.
>
> > It is counterintuitive that more iid data (larger $\alpha$) results in worse performance than non-iid data. Although the authors claim it is due to a lack of sufficient parameter tuning, it is not convincing. More careful experiments are desired.
>
> We appreciate the reviewer’s observation regarding the counterintuitive result that more IID data (larger $\alpha$) leads to worse performance than non-IID data. Our initial claim was based on the higher standard deviations observed in Fig 3(a), which indicated variability in performance rather than a consistent trend.  For more clarity, we did perform additional experiments to substantiate our claims.
>
> | α   | Final         |
> |-----|---------------|
> | 0.1 | 64.71±0.53    |
> | 0.2 | 64.32±1.25    |
> | 0.3 | 63.74±0.39    |
> | 0.4 | 64.67±0.35    |
> | 0.5 | 62.87±0.20    |
> | 0.6 | 64.58±0.28    |
> | 0.7 | 64.00±0.77    |
> | 0.8 | 64.23±0.22    |
> | 0.9 | 65.21±0.54    |
> | 1   | 66.05±0.20    |
>
> The revised table provided above demonstrates that the performance across different $\alpha$ values exhibits nuanced patterns, with higher  $\alpha$ values (more IID data) generally improving the average accuracy [[1](https://arxiv.org/pdf/2502.00182v1), [2](https://fl-icml.github.io/2021/papers/FL-ICML21_paper_59.pdf)]. This reinforces the idea that sufficient parameter tuning and experimental rigor are necessary to fully capture the dynamics of Fed-GNODEFormer in varying data distributions. We will expand the discussion in the manuscript to include this revised analysis and address any inconsistencies more comprehensively.
>
> [1] Seo, Jungwon, Ferhat Ozgur Catak, and Chunming Rong. "Understanding Federated Learning from IID to Non-IID dataset: An Experimental Study." arXiv preprint arXiv:2502.00182 (2025).
>
> [2] Xie, Han, et al. "Federated graph classification over non-iid graphs." Advances in neural information processing systems 34 (2021): 18839-18852.
>
> > Minor Changes
>
> We thank the reviewer for highlighting these points. We will replace "alpha" with the correct symbol, ensure consistent usage of \citet{} and \citep{}, and revise the notations to eliminate overloading and improve clarity. All symbols and indexing will be redefined for better readability in the revised manuscript.

---

### Review · Reviewer_v6xZ · 2025-02-04

**Summary Of Contributions:**

This paper proposed a novel federated learning method called Fed-GNODEFormer for training GNNs on non-IID graph data. By integrating spectral GNNs and Neural Ordinary Differential Equations, this method aims to better capture complex relationships in graph structures and demonstrates superior performance on both homophilic and heterophilic graphs. The authors provided a detailed description of the model architecture, including the centralized GNODEFormer and its extension to the federated learning scenario, Fed-GNODEFormer. Extensive experiments validate the effectiveness of this approach under various data distribution conditions. Overall, this research is innovative and practical in the field of federated graph learning, offering a new perspective for handling complex graph data.

**Audience:**

Yes

**Claims And Evidence:**

Yes

**Requested Changes:**

See  weaknesses

**Strengths And Weaknesses:**

### Strengths
1. **Innovative Architecture**: The combination of spectral GNNs and Neural ODEs to form the GNODEFormer architecture, and its extension to the federated learning scenario as Fed-GNODEFormer, is novel. This integration allows for better capturing of complex relationships and dynamic changes between nodes in graph data, especially under non-IID conditions, bringing new technical ideas to federated graph learning.
2. **Strong Capability for Non-IID Data**: The experimental results show that this method outperforms existing methods under extremely non-IID conditions (e.g., α=0.01), effectively addressing the issue of uneven data distribution across different clients. This is significant for many real-world applications such as social network analysis and recommendation systems, where data is often highly heterogeneous and unevenly distributed.
3. **Wide Applicability**: Fed-GNODEFormer not only performs well on non-IID data but also achieves comparable performance on IID data, demonstrating its adaptability and robustness across different data distributions. Moreover, its good performance on both homophilic and heterophilic graphs further proves its universality in diverse graph structures.
4. **Privacy and Communication Optimization**: The authors emphasize the design of Fed-GNODEFormer in terms of data privacy protection and communication cost optimization, which are crucial for practical applications of federated learning. In real-world scenarios, data privacy and communication efficiency are often key factors restricting the development of federated learning, and Fed-GNODEFormer achieves a good balance between them, making it closer to the requirements of practical applications.

###  Weaknesses
1. **Computational Efficiency**: Despite its great performance, Fed-GNODEFormer suffers from relatively low computational efficiency, especially when dealing with larger heterophilic graph datasets. For example, on the Squirrel and Actor datasets, the time required for each local training epoch is quite long when using the fourth-order Runge-Kutta method (RK-4), which may limit its application in large-scale or resource-constrained environments.
2. **Model Complexity and Scalability**: As the scale of graph data increases and the graph structure becomes more complex, the complexity of Fed-GNODEFormer may rise further. The paper does not provide detailed discussions on how to effectively scale this method to large-scale graph data, which may affect its applicability in industrial-level applications.
3. **Hyperparameter Selection and Tuning**: The high-order Runge-Kutta methods in Neural ODEs involve multiple hyperparameters (such as step size and order), which significantly affect the model's performance and computational efficiency. However, the paper provides relatively little discussion on these hyperparameters and lacks systematic strategies for hyperparameter selection and tuning. addressed.

---

> ### Author Response · Authors · 2025-02-09
> **Response to Reviewer v6xZ**
>
> We thank the reviewer for their comments on our work. We provide a detailed rebuttal for the same below.
>
> > **Computational Efficiency**: Despite its great performance, Fed-GNODEFormer suffers from relatively low computational efficiency, especially when dealing with larger heterophilic graph datasets. For example, on the Squirrel and Actor datasets, the time required for each local training epoch is quite long when using the fourth-order Runge-Kutta method (RK-4), which may limit its application in large-scale or resource-constrained environments.
>
> We thank the reviewer for their question. Our method inherits standard GNN scaling challenges, such as memory and computation overhead. Existing solutions like sampling or sparsification can address these issues and are areas for future work. Section A.3 analyzes computational and communication overhead in FL setups, comparing RK-2 and RK-4 solvers. Table 8 shows that RK-4 achieves better accuracy on complex heterophilic graphs but with 1.65x higher computational cost (e.g., on the Photo dataset), while RK-2 is faster and suitable for homophilic or resource-constrained cases. Additionally, results added in tZrz's response show that our model scales well on larger datasets, balancing computational demands with performance.
>
> > **Model Complexity and Scalability**: As the scale of graph data increases and the graph structure becomes more complex, the complexity of Fed-GNODEFormer may rise further. The paper does not provide detailed discussions on how to effectively scale this method to large-scale graph data, which may affect its applicability in industrial-level applications.
>
> We thank the reviewer for their observation. Our response to tZrz addresses this concern, including a detailed discussion in Section A.3 on the computational trade-offs of RK-2 and RK-4 solvers and their impact on performance.
>
> > **Hyperparameter Selection and Tuning**: The high-order Runge-Kutta methods in Neural ODEs involve multiple hyperparameters (such as step size and order), which significantly affect the model's performance and computational efficiency. However, the paper provides relatively little discussion on these hyperparameters and lacks systematic strategies for hyperparameter selection and tuning. addressed.
>
> We thank the reviewer for their feedback. As noted in our response to tZrz, we employed standard hyperparameter tuning strategies, such as grid search over learning rates, hidden dimensions, and ODE solver steps. Additionally, as detailed in our response to 87kj, we conducted an ablation study to analyze the impact of these choices. These details will be included in the revised manuscript, with further information provided in the appendix.

---

### Decision · Action_Editor_J7Sn · 2025-03-13

**Recommendation:** Accept with minor revision

**Comment:**

The paper presents a novel federated learning approach for GNNs using spectral methods and neural ODEs. The results are promising, especially for non-IID heterophilic graphs. Minor revisions are needed to clarify the experimental setup and provide more detailed comparisons with existing methods.

**Audience:**

The work is relevant to TMLR’s audience, addressing a timely problem in federated learning and graph neural networks.

**Claims And Evidence:**

The claims are supported by clear and convincing evidence. The proposed method effectively handles non-IID data and shows strong performance on both homophilic and heterophilic graphs.